:ͻ: PLOS | ONE

# Variation in the LRR region of Pi54 protein alters its interaction with the AvrPi54 protein revealed by *in silico* analysis

**Chiranjib Sarkar**[1,2,3], **Banita Kumari Saklani**[3,4], **Pankaj Kumar Singh**[3,5], **Ravi Kumar Asthana**[4], **Tilak Raj Sharma**[3,5]*

**1** ICAR-Indian Agricultural Research Institute, New Delhi, India, **2** ICAR-Indian Agricultural Statistics Research Institute, New Delhi, India, **3** ICAR-National Research Centre on Plant Biotechnology, New Delhi, India, **4** Banaras Hindu University, Varanasi, Uttar Pradesh, India, **5** National Agri-Food Biotechnology Institute, Mohali, Punjab, India

* trsharma@nabi.res.in

**Data Availability Statement:** All relevant data are within the paper and its Supporting Information files.

## Abstract

Rice blast, caused by the ascomycete fungus *Magnaporthe oryzae* is a destructive disease of rice and responsible for causing extensive damage to the crop. *Pi54*, a dominant blast resistance gene cloned from rice line Tetep, imparts a broad spectrum resistance against various *M. oryzae* isolates. Many of its alleles have been explored from wild *Oryza* species and landraces whose sequences are available in the public domain. Its cognate effector gene *AvrPi54* has also been cloned from *M. oryzae*. Complying with the Flor's gene-for-gene system, Pi54 protein interacts with AvrPi54 protein following fungal invasion leading to the resistance responses in rice cell that prevents the disease development. In the present study *Pi54* alleles from 72 rice lines were used to understand the interaction of Pi54 (R) proteins with AvrPi54 (Avr) protein. The physiochemical properties of these proteins varied due to the nucleotide level polymorphism. The *ab initio* tertiary structures of these R- and Avr- proteins were generated and subjected to the *in silico* interaction. In this interaction, the residues in the LRR region of R- proteins were shown to interact with the Avr protein. These *R* proteins were found to have variable strengths of binding due to the differential spatial arrangements of their amino acid residues. Additionally, molecular dynamic simulations were performed for the protein pairs that showed stronger interaction than Pi54$^{tetep}$ (original Pi54 from Tetep) protein. We found these proteins were forming h-bond during simulation which indicated an effective binding. The root mean square deviation values and potential energy values were stable during simulation which validated the docking results. From the interaction studies and the molecular dynamics simulations, we concluded that the AvrPi54 protein interacts directly with the resistant Pi54 proteins through the LRR region of Pi54 proteins. Some of the Pi54 proteins from the landraces namely Casebatta, Tadukan, Varun dhan, Govind, Acharmita, HPR-2083, Budda, Jatto, MTU-4870, Dobeja-1, CN-1789, Indira sona, Kulanji pille and Motebangarkaddi cultivars show stronger binding with the AvrPi54 protein, thus these alleles can be effectively used for the rice blast resistance breeding program in future.

**Funding:** TRS is thankful to the Department of Biotechnology, Govt. of India, for financial help and Department of Science and Technology, Govt of India, for JC Bose National Fellowship. CS is thankful to IARI, New Delhi, India, for the financial assistance and BKS is thankful to CSIR, New Delhi, India, for research grant. The funders had no role in study design, data collection and analysis, decision to publish, or preparation of the manuscript.

**Competing interests:** The authors have declared that no competing interests exist.

# Introduction

Rice is the widely consumed staple food throughout the globe. Rice is predominantly cultivated in Asia and it is the source of 23% of the calories consumed by the global human population [1]. The crop is vulnerable to a number of biotic and abiotic stresses. High losses incurred due to various diseases can threaten the global food security. Rice blast, caused by the fungus *Magnaporthe oryzae*, causes a big loss to the net production of rice [2]. Lot of research is being undertaken to explore the genomics, host-pathogen interactions, mechanism of development of disease and breeding strategies with an aim to establish effective disease management strategy. Whole genome sequences for both the organisms are available in public domain which have accelerated the efforts to identify and characterize the determinants of blast disease, both in rice and the fungus. Some determinants of the disease development are avirulence (*Avr*) genes in the pathogen which condition pathogenicity and resistance (*R*) genes in the host which condition the defense reaction in response to the pathogen invasion. Many *R* genes from rice and *Avr* genes from *M. oryzae* have been identified and characterized. Nearly 350 QTLs for resistance to rice blast and 101 *R* genes have been identified out of which 23 *R* genes have been molecularly characterized [3,4,5]. A total of 25 *Avr* genes of *M. oryzae* have been genetically mapped out of which 11 *Avr* genes have also been cloned and characterised [6]. The pathogenic races of *M. oryzae* which carry the dominant *Avr* gene are unable to develop disease in certain cultivars of the host species which carry the dominant cognate *R* gene. Such host plants develop defense responses following the fungal infection and restrict the disease development. These *Avr* and *R* gene follow the gene-for-gene hypothesis [7]. The hypothesis implies that the Avr proteins can be recognized directly in those cultivars of the host species which has functional protein coded by corresponding *R* genes. Such cultivars do not develop the disease. The R-Avr interaction triggers the hypersensitive cell death in the plant which kills the infected cell, thereby checking the invasion of the pathogen to non-infected cells. The interaction between the R- and Avr- proteins can even be direct through some modified proteins of the host [8, 9].

The exploitation of host plant resistance is one of the most economical and environmentally safe approaches to develop resistance against blast disease [10]. Efforts are going on for identification and the deployment of the *R* genes to develop resistant cultivars against rice blast worldwide. A dominant *R* gene, *Pi54* has been cloned from *indica* rice line 'Tetep' and validated to impart wide spectrum resistance in rice against diverse *M. oryzae* strains [11,12,13]. It is responsible for activation of defence response genes in response to the pathogen attack [14]. It contains a zinc finger domain and an LRR domain [11,15]. The C-terminal LRR domain of the *R* genes is involved in ligand recognition and binding [16]. This interaction activates the host defense mechanism via signal cascade leading to activation of the genes involved in hypersensitive resistance response [3]. LRR region exhibits more variations than other regions of the gene [17]. Orthologues of *Pi54* named as *Pi54of* and *Pi54rh* that confer high degree of resistance to *M. oryzae* were cloned from wild species of rice [18,19]. The *in silico* protein modelling and molecular docking between Pi54 protein and candidate effector proteins from *M. oryzae* have been used as a strategy to find the probable avirulent *AvrPi54* gene by analyzing the interaction at the LRR domain of the Pi54 protein. *AvrPi54* has then been cloned and characterised [20]. The molecular docking was also used to display interaction of *Pi54of* protein with *Avr-Pi54* through STI1 and RhoGEF domains which are components of the rice defensome complex [19].

The Pi54 and AvrPi54 interactions are essential to understand the mechanism of blast disease development. There are many allelic variants of the Pi54 resistance gene explored from various landraces and are reported to have unique polymorphic patterns at nucleotide

level [16]. The sequence polymorphism in these alleles can result in structural variation at the protein level which can alter the interacting surface and impact the interaction potential with AvrPi54 protein. Our present study has analyzed the impact of nucleotide level polymorphism among Pi54 alleles and on their properties, structure and interaction with AvrPi54 protein. We have determined structures of 72 allelic Pi54 proteins and studied their interaction with the AvrPi54 protein by molecular docking using *in silico* tools, aimed to find more suitable alleles which can be used in breeding programs to develop blast resistance in rice. The online available bioinformatics tools for protein modelling and interaction have been used in this study [21]. Thus it saves the time and cost of the experimental structure determinations by X-Ray Crystallography and nuclear magnetic resonance. The most efficient docking algorithms often produce models with atomic-level accuracy. Molecular docking is widely used in drug discovery as it is one of the most reliable methods for the prediction of the interaction between two molecules [22]. It allows the assessment of the key residues located at the active site of the target molecule that participates in interaction with the ligand. The approach has been extended to the R-Avr interaction analysis. [19,20,23]. The alleles whose protein products show more interaction with the AvrPi54 protein can further be examined experimentally and deployed in the development of more resistant cultivars.

## Materials and methods

### Sequence retrieval

The nucleotide sequence of the *AvrPi54* gene with the accession number HF545677 was retrieved from European Nucleotide Archive (ENA) of European Molecular Biology Laboratory (EMBL) Nucleotide Sequence Database. The nucleotide sequence of the blast resistance gene *Pi54* (*Pi54^tetep^*), (Accession no. AY914077) was retrieved from NCBI gene database (www.ncbi.nlm.nih.gov/). The nucleotide sequence of 72 alleles of *Pi54* from different land races of rice has been retrieved from the EMBL database which is listed in S1 Table.

### Domain identification in Pi54 proteins

The amino acid sequence of all the *Pi54* alleles were deduced from the nucleotide sequences using the web based tool FGENESH (http://linux1.softberry.com/berry). The monocot plants (Corn, Rice wheat, Barley) were selected under organism specific gene-finding parameters. The LRR domain was identified in these allelic proteins by aligning with the already predicted LRR domain of the Pi54 protein originally cloned from rice variety Tetep as described earlier [11].

### Multiple sequence alignment of the LRR region

The nucleotide sequences of the LRR region of *Pi54* alleles were aligned with the nucleotide sequence of *Pi54* gene from Tetep one by one using the online web-based tool Clustal omega (http://www.ebi.ac.uk/Tools/msa/clustalo/). The output file of the Clustal omega was saved in FASTA format and analysed using DNASP (DNA sequence polymorphism) standalone software [24] to identify SNPs and InDels. Amino acid substitutions and frame shifts in these Pi54 proteins were determined by the Clustal omega software (http://www.ebi.ac.uk/Tools/msa/clustalo/). The output of the Clustal omega tool was viewed in JalView [25] to identify amino acid substitution and conserved regions in the alignment.

## Determination of physiochemical properties of various Pi54 proteins

The primary structure of the Pi54 proteins was studied using ProtParam tool (http://web.expasy.org/protparam/) of Expasy Server and molecular weight, isoelectric point, instability index, aliphatic index, and grand average hydropathy (GRAVY) were computed.

## Tertiary structure (3D) prediction and modelling of Pi54 proteins

All the protein sequences of the Pi54 alleles used in this study showed less than 10% identity in similarity search against Protein Data Bank [26] (http://www.rcsb.org/pdb/home/home.do) using the BLAST tool [27]. Therefore, their tertiary structure (3D) prediction was done with *ab initio* based protein modelling procedure using the web-based server I-TASSER (Iterative Threading Assembly Refinement) (http://zhanglab.ccmb.med.umich.edu/I-TASSER/). Each predicted protein structure was visualized in RasMol visualization tool [28] and the numbers of different secondary structures like alpha-helix, beta-sheet, turns, coil and total numbers of hydrogen bonds etc. were calculated. The 3D structures of Pi54 proteins were assessed by Ramachandran plot [29] in Discovery Studio 2.0 (Accelrys Life Science Tool). Refinement of structure was done for those models that showed residues below the expected value (~98%) in favourable region in Ramachandran plot using the ModRefiner server (http://zhanglab.ccmb.med.umich.edu/ModRefiner/) and the residues in favourable region were increased. The energy minimization of the refined models was performed by CHARMM 19 (Chemistry at HARvard Macromolecular Mechanics) Force Field [30]. These procedures of structure refinement and energy minimization were iterated till the optimized structures were obtained. Similar strategy was employed to get the tertiary structure of mature AvrPi54 protein after cleaving the signal peptide region of 19 amino acids. The signal peptide sequence was predicted with TargetP tool (http://www.cbs.dtu.dk/services/TargetP/).

## Quantitative similarity assessment of protein structures

The similarity of the various Pi54 protein structures were compared with the Pi54$^{tetep}$ protein using the TM-align tool (http://zhanglab.ccmb.med.umich.edu/TM-align/). This tool performs the structural comparison and residue-to-residue alignment. The optimal superposition of two protein structure gives the TM-score value and RMSD value. The TM-score was calculated for the residue pairs of the two proteins within 0.5 Å atomic distances.

## Molecular docking of Pi54 and AvrPi54 proteins

Each Pi54 protein and AvrPi54 protein pair was subjected to molecular docking by the stand-alone Z-Dock software available in the Discovery Studio 2.0 (Accelrys Life Science Tool). This method performs calculation of docked protein poses, filtering of docked protein poses, re-rank docked protein poses and cluster docked protein poses and calculation of cluster density. The docking was first performed between the Pi54$^{tetep}$ protein and the AvrPi54 protein. The result of the interaction analysis of the docking of Pi54$^{tetep}$ protein and the AvrPi54 protein was used as a control and same parameters were used for other docking analysis (Table 1). The best docked pose for each pair was thus determined and used for further analysis.

## Calculation of binding energy

Depending on the docking results of Pi54 and AvrPi54 proteins the binding energy of the interaction were calculated for the interacted pairs. The binding energy depends on the interacting residues of the two proteins and the atoms participating in the interaction.

**Table 1. The parameters used for performing docking in Z-Dock software.**

| Parameters | Value |
|---|---|
| Angular step size | 6 |
| Distance cut-off | 10.0 |
| ZRank | True |
| Zrank Top poses | 10 |
| Clustering Top poses | 10 |
| Clustering RMSD cut-off | 10.0 |
| Clustering Interface cut-off | 10.0 |
| Maximum number of clusters | 2 |
| Parallel processing | False |
| Parallel processing server order | True |
| Use electrostatic and desolvation energy | True |

The energy difference was calculated using the equation:

$$QE = Ecomplex - Eligand - Eprotein \ (QE \ is \ the \ ligand \ binding \ energy)$$

### Molecular dynamic simulation

In order to assess the reliability of the docking results and to understand their stability, molecular dynamics simulations was performed for the chosen poses of a few pairs of proteins under our study from the docking results by using GROMACS version 2018.1 [31] with the force field as GROMOS96 54a7 force field [32]. The docked structures with minimum binding energy and maximum interaction values were chosen and executed for 100 ps long MD simulations, and conformations were saved at 0.001 ps intervals. The proteins structures of each pair were solvated, minimized and equilibrated. The solvation was done with spc216 water model in a cubic box ($10.4 \times 10.4 \times 10.4$ nm$^3$) and the Counter-ion (Na$^+$) was included to counterbalance the solvated system. To minimise the steric hindrances in the solvated system of protein–ligand complex, energy minimization was done using the steepest algorithm up to a maximum 50,000 steps or until the maximum force (Fmax) did not exceed the default threshold of 1000 kJ mol$^{-1}$ nm$^{-1}$. The system was first equilibrated using NVT ensemble followed by NPT ensemble for 50,000 steps (100 ps) at 300 K temperature and 1 atm pressure. The molecular dynamic simulations were carried out for 2 ns long and the Root Mean Square Deviation (RMSD), Root Mean Square Fluctuation (RMSF), hydrogen bonds and energy plots were generated.

## Results

### Physio-chemical properties of Pi54 proteins

The amino acid sequences for all the alleles were analyzed and compared to the *Pi54$^{tetep}$*. The size of the Pi54 proteins under study varied from 173 amino acids to 486 amino acids while the Pi54$^{tetep}$ protein is 330 amino acids long. Their molecular weights were quite variable due to the change in number of amino acid residues in the proteins. These proteins were found to have high content of some amino acids such as leucine, glutamic acid and cysteine. Maximum percentage of Leucine in these proteins was nearly 17% (S1 Fig). Their physio-chemical properties like theoretical pI, GRAVY, aliphatic index and molecular weight (Table 2) showed variation from the Pi54$^{tetep}$ protein. All the proteins except Dobeja-1, ND 118 and Samleshwari

**Table 2. The physio-chemical properties of the Pi54 proteins.**

| Rice lines | Theoretical pI | Molecular weight (kDa) | Leucine % | GRAVY Index | Aliphatic index |
|---|---|---|---|---|---|
| Tetep | 5.00 | 37.30 | 16.70 | -0.05 | 104.00 |
| Acharmita | 6.31 | 27.38 | 17.50 | 0.14 | 105.62 |
| Basmati 386 | 5.12 | 51.92 | 17.40 | 0.01 | 105.57 |
| Belgaum basmati | 5.12 | 43.09 | 17.30 | 0.08 | 108.24 |
| Bidarlocal-2 | 4.91 | 21.54 | 16.60 | -0.09 | 102.59 |
| Budda | 6.31 | 31.84 | 16.20 | 0.14 | 101.69 |
| Casbatta | 5.12 | 19.39 | 16.80 | -0.20 | 98.67 |
| Chiti zhini | 5.39 | 45.38 | 17.50 | 0.06 | 107.21 |
| CN-1789 | 5.18 | 30.63 | 17.10 | 0.06 | 106.95 |
| CSR 10 | 5.34 | 45.26 | 17.20 | 0.04 | 104.31 |
| CSR-60 | 5.12 | 43.09 | 17.30 | 0.08 | 108.24 |
| Dobeja-1 | 8.35 | 32.25 | 17.00 | 0.25 | 105.41 |
| Gonrra bhog | 5.18 | 43.45 | 16.30 | 0.13 | 107.19 |
| Govind | 5.59 | 41.96 | 15.50 | 0.14 | 101.17 |
| Gowrisanna | 5.06 | 39.22 | 14.50 | 0.07 | 102.56 |
| Himalya 799 | 5.55 | 50.02 | 17.20 | 0.06 | 105.87 |
| HLR-108 | 5.12 | 43.10 | 17.30 | 0.08 | 108.24 |
| HLR-142 | 5.12 | 43.10 | 17.30 | 0.08 | 108.24 |
| HPR 2083 | 5.07 | 40.20 | 16.90 | 0.02 | 105.46 |
| HPR-2178 | 5.25 | 28.12 | 17.30 | -0.02 | 107.34 |
| HR-12 | 4.82 | 23.31 | 16.30 | -0.11 | 103.59 |
| IC356437 | 5.12 | 43.10 | 17.30 | 0.08 | 108.24 |
| Indira sona | 5.51 | 50.13 | 17.20 | 0.05 | 104.33 |
| Indrayani | 5.93 | 47.36 | 16.70 | -0.07 | 102.60 |
| INRC 779 | 5.18 | 30.63 | 17.10 | 0.06 | 106.95 |
| IR 64 | 4.84 | 21.63 | 16.70 | -0.08 | 102.59 |
| IRAT-144 | 5.07 | 40.37 | 17.50 | 0.05 | 108.20 |
| IRBB 55 | 5.45 | 54.30 | 17.70 | -0.02 | 104.81 |
| IRBB-13 | 5.12 | 43.09 | 17.30 | 0.08 | 108.24 |
| IRBB-4 | 5.16 | 54.85 | 16.90 | -0.01 | 103.13 |
| Jatto | 5.64 | 49.98 | 17.10 | 0.01 | 103.05 |
| Kari kantiga | 5.12 | 43.10 | 17.30 | 0.08 | 108.24 |
| Kariya | 5.12 | 43.10 | 17.3 | 0.08 | 108.24 |
| Kasturi | 5.33 | 47.78 | 17.10 | 0.06 | 106.06 |
| Kavali kannu | 6.31 | 27.39 | 17.50 | 0.14 | 105.62 |
| Kulanji pille | 5.44 | 43.73 | 15.00 | -0.01 | 98.77 |
| LD-43 (HLR-144) | 5.24 | 42.45 | 17.60 | 0.01 | 107.07 |
| Lalnakanda | 5.12 | 43.10 | 17.30 | 0.08 | 108.24 |
| Mahamaya | 5.34 | 45.26 | 17.20 | 0.04 | 104.31 |
| Malviya dhan | 5.38 | 39.15 | 17.40 | 0.04 | 104.30 |
| Mesebatta | 5.18 | 30.63 | 17.10 | 0.06 | 106.95 |
| Mote bangarkaddi | 5.18 | 30.63 | 17.10 | 0.06 | 106.95 |
| MTU 4870 | 5.93 | 48.82 | 14.60 | 0.06 | 100.00 |
| MTU-1061 | 5.09 | 43.11 | 16.50 | 0.04 | 107.98 |
| ND 118 | 8.33 | 35.21 | 17.40 | 0.10 | 101.58 |
| Orugallu | 5.13 | 40.26 | 16.90 | 0.02 | 105.46 |

*(Continued)*

**Table 2.** (Continued)

| Rice lines | Theoretical pI | Molecular weight (kDa) | Leucine % | GRAVY Index | Aliphatic index |
|---|---|---|---|---|---|
| Pant sankar dhan 1 | 5.12 | 43.10 | 17.30 | 0.08 | 108.24 |
| Pant sankar dhan 17 | 5.67 | 40.32 | 18.00 | -0.03 | 104.63 |
| Parijat | 5.32 | 45.36 | 17.70 | 0.07 | 108.18 |
| Parimala kalvi | 5.39 | 45.38 | 17.50 | 0.06 | 107.21 |
| PR 118 | 4.98 | 21.58 | 16.60 | -0.10 | 102.07 |
| Pusa 33 | 5.48 | 41.56 | 17.40 | 0.04 | 106.78 |
| Pusa basmati 1 | 5.27 | 45.32 | 17.20 | 0.03 | 104.06 |
| Pusa Sugandh 3 | 5.12 | 43.10 | 17.30 | 0.08 | 108.24 |
| Pusa Sugandh 4 | 5.12 | 43.10 | 17.30 | 0.08 | 108.24 |
| Ram Jawain 100 | 5.14 | 40.32 | 16.90 | 0.04 | 106.85 |
| Ranbir basmati | 5.12 | 43.10 | 17.30 | 0.08 | 108.24 |
| Sadabahar | 5.32 | 45.36 | 17.70 | 0.07 | 108.18 |
| Samleshwari | 8.64 | 27.12 | 17.50 | 0.12 | 105.62 |
| Sanna mullare | 5.18 | 30.63 | 17.10 | 0.06 | 106.95 |
| Sathia -2 | 5.24 | 43.22 | 17.40 | 0.03 | 106.61 |
| Satti | 5.34 | 45.26 | 17.20 | 0.04 | 104.31 |
| Shiva | 5.63 | 43.06 | 15.10 | 0.09 | 94.63 |
| Superbasmati | 5.12 | 43.10 | 17.30 | 0.08 | 108.24 |
| Suphala | 5.32 | 45.36 | 17.70 | 0.07 | 108.18 |
| T23 | 5.33 | 47.66 | 17.10 | 0.08 | 106.31 |
| Tadukan | 4.78 | 21.54 | 16.60 | -0.09 | 102.59 |
| Taipei-309 | 5.18 | 30.63 | 17.10 | 0.06 | 106.95 |
| Thule ate | 5.63 | 43.06 | 15.10 | 0.09 | 94.63 |
| Tilak chandan | 5.12 | 43.10 | 17.30 | 0.08 | 108.24 |
| Tiyun | 5.69 | 36.08 | 16.90 | -0.01 | 104.20 |
| V L Dhan | 6.31 | 27.38 | 17.50 | 0.14 | 105.62 |
| Vanasurya | 5.18 | 30.63 | 17.10 | 0.06 | 106.95 |
| Varalu | 5.32 | 45.36 | 17.70 | 0.07 | 108.18 |
| Varun dhan | 5.19 | 35.51 | 16.00 | -0.07 | 101.57 |

were acidic as they had pI values in the acidic range. The aliphatic index of a protein is an indicator of the relative volume occupied by aliphatic side chains of amino acids such as alanine, valine, leucine, and isoleucine. It measures thermal stability of the proteins [33]. Aliphatic index was high for most of the proteins. The GRAVY value of the Pi54$^{tetep}$ protein was negative, *i.e.* -0.054 which indicates its good affinity for water. The GRAVY value of proteins in this study had both negative and positive values. GRAVY values for all the *Pi54* proteins were observed to be greater than Pi54$^{tetep}$ protein except the alleles derived from Bidarlocal-2, Casbatta, Indrayani, IR-64, PR-118, Tadukan, Varun dhan and HR-12. The GRAVY values showed very large variation in case of the proteins like Basmati-386, HPR-2083, HPR-2178, IRBB-55, IRBB-4, Jatto, Kulanji Pille, LD-43 and Tiyun.

## Analysis of LRR region in Pi54 proteins

A stretch of 45 amino acids from 267 to 311 amino acids has been predicted as the LRR region in Pi54$^{tetep}$ protein [11]. The LRR region in all the Pi54 proteins was predicted by their sequence alignment to the Pi54$^{tetep}$ protein. The amino acid sequences of the Pi54 proteins

Fig 1. Alignment of LRR region of the Pi54 proteins. (A) shows conservation and (B) shows variation in LRR region using few representative proteins. Pi54tetep protein used as control.

from some cultivars showed high conservation in LRR region (Fig 1A), while in some others, LRR region contained more number of substitutions (Fig 1B). The average leucine percentage and the number and arrangements of alpha-helix, beta-sheets and turns in some of the Pi54 proteins having conserved LRR region were similar to that of Pi54tetep protein, whereas these protein features varied for the other proteins having diverse LRR region (Fig 2).

## Tertiary structure prediction and modelling of the Pi54 proteins

The three-dimensional (3D) structures of all the Pi54 proteins are given in Fig 3 and S2 Fig. The Pi54tetep protein showed a typical horseshoes shaped structure but such geometry was not seen in all the Pi54 proteins. The horseshoe shaped structures were obtained for 36 proteins. Each protein molecule showed a motif having a curved region lined with the parallel beta strands on inner side and all the alpha helices on the one side of the beta sheet. The α-helices and β-sheets folded into tertiary structure and they were stabilized by hydrogen bonds. In case of the Pi54tetep protein the number of helices, strands and turns were predicted as 9, 17 and 53, respectively, and total 244 H-bonds were also identified in this protein. The numbers of helices, strands and turns varied in different Pi54 proteins (Table 3). The potential energy, vander

| Rice lines | LRR location (amino acids) | Secondary structure in LRR region |
|---|---|---|
| Pi54 | 267-330 | C C C C C E E E E E C C C C C C E C C H H H C C C C C C C E E E E E C C C C C E E C C C |
| Suphala | 313-357 | C C C C C E E E E E C C C C C C E C C H H H C C C C C C C E E E E E C C C C C E E C C C |
| Casebatta | 130-157 | C C C C C E E E E E C C C C C C E C C H H H C C C C C C C E E E E E C C C C C E E C C C |
| HR-12 | 146-190 | C C C C C E E E E E C C C C C C C C C C C C C C C C C C C E E E E E C C C C C C E C C C |
| Kavali kannau | 112-156 | C C C C C E E E E E C C C C C C C C C H H H C C C C C C C E E E E E C C C C C E E C C C |
| Lalnakanda | 293-337 | C C C C C E E E E C C C C C C C C C C C C C C C C C C C C E E E E E C C C C C C E C C C |
| Mahamaya | 313-357 | C C C C C E E E E E C C C C C C E C C C C C C C C C C C C E E E E E C C C C C E E E C C |
| Malviya dhan | 215-259 | C C C C C E E E E E C C C C C C C C C C C C C C C C C C C E E E E E C C C C C C C C C C |
| Masebatta | 181-225 | C C C C C E E E E E C C C C C C C C C E E E C C C C C C C E E E E E C C C C C E E C C C |
| Parijat | 313-357 | C C C C C E E E E C C C C C C C C C C H H H C C C C C C E E E E E E C C C C C C E C C C |
| Ram jawain 100 | 267-311 | C C C C C E E E E E C C C C C C C C C C E E C C C C C C C E C C E E C C C C C E E C C C |
| Sathia-2 | 313-357 | C C C C C E E E E E C C C C C C C C C E C E C C C C C C C E E E E E C E E E C E E C C C |
| Shiva | 210-254 | C C C C C E E E E E C C C C C C C C C C H H E C C C C C C C E C C E E C C C C C E E C C C |
| Super Basmati | 293-337 | C C C C C E E E E E C C C C C C E C C C C C C C C C C C C E E E E E C C C C C E E E C C |
| T23 | 293-337 | C C C C C C C C E E C C C C C C C E C C C C C C C C C C C E E E E E C C C C C E E E C C |
| TP-309 | 181-225 | C C C C C E E E E E C C C C C C E C C C C C C C C C C C C E E E E E C C C C C E C C C C |

**Fig 2. Comparison of secondary structural elements in the LRR region of the Pi54 proteins to the Pi54[tetep] protein.** Secondary structural elements in some of the Pi54 proteins were conserved whereas but varied in few other proteins having diverse LRR region when compared to Pi54[tetep] protein. C- coils, E- β-sheet, H- helices.

waals energy, electrostatic energy and sum of all the total energy, *i.e.* global free minimum energy were calculated for all Pi54 proteins (Table 4). The total energy of Pi54[tetep] protein is -37033.6752kcal/mol. The global free minimum energy of the Pi54 proteins showed that some of these proteins from rice lines like Kasturi, Kulanji pille, HLR-144, Mahamaya, ND 118 etc. had lesser energy than the Pi54[tetep].

The quantitative assessment of similarity of 3D structures was done by determining TM-score and RMSD values for each pair of Pi54 protein and the Pi54[tetep] protein. The data is given in Table 5. The superposition of the two proteins was generated by the residue-to-residue alignments. The TM-align tool gives scores between 0 and 1, the TM-score < 0.2 indicated no similarity between two structures and a TM-score > 0.5 means the structures share the same fold. The Pi54 protein from the cultivar Orugallu had TM-score 0.6158 hence shared same fold as that of the Pi54[tetep] protein and also had highest number (183) of identical residues. Proteins from Acharmita, V L Dhan and Shiva showed no similarity to the Pi54[tetep] protein as their TM-scores were < 0.2. The proteins of HPR-2083 and IRAT-144 contained 124

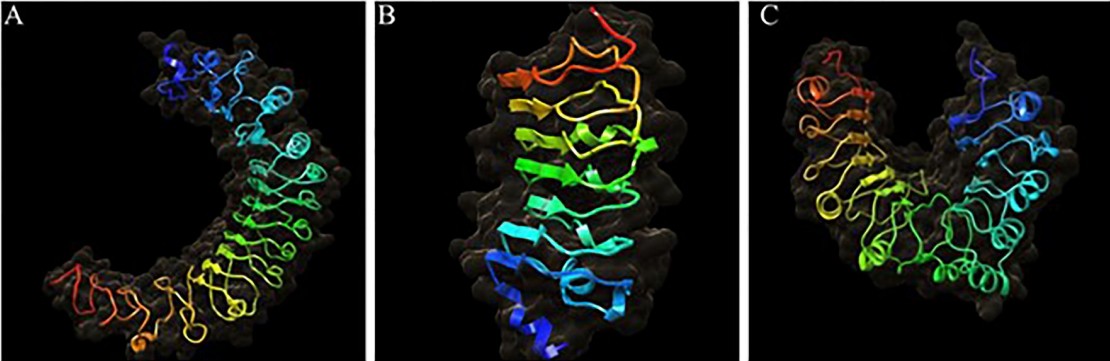

**Fig 3. Three dimensional structures of Pi54 proteins.** Some of R proteins showed a typical horseshoes shaped structure but such geometry was not seen in all the Pi54 proteins. 3D structures from rice lines, A-Tetep, B- Casbatta, C- HPR– 2083.

**Table 3. Secondary structures of the Pi54 proteins.**

| Rice lines | Helices | Strands | Turns | Rice lines | Helices | Strands | Turns |
|---|---|---|---|---|---|---|---|
| Tetep | 9 | 17 | 53 | Lalnakanda | 14 | 18 | 66 |
| Acharmita | 3 | 17 | 29 | Mahamaya | 11 | 26 | 65 |
| Basmati 386 | 14 | 30 | 85 | Malviya dhan | 3 | 21 | 61 |
| Belgaum basmati | 14 | 18 | 66 | Mesebatta | 6 | 15 | 39 |
| Bidarlocal-2 | 4 | 15 | 46 | Mote bangarkaddi | 5 | 24 | 41 |
| Budda | 11 | 23 | 40 | MTU 4870 | 10 | 25 | 45 |
| Casbatta | 8 | 13 | 25 | MTU-1061 | 12 | 19 | 58 |
| Chiti zhini | 14 | 22 | 66 | ND 118 | 13 | 18 | 42 |
| CN-1789 | 8 | 19 | 42 | Orugallu | 9 | 26 | 61 |
| CSR 10 | 13 | 34 | 58 | Pant sankar dhan 1 | 13 | 34 | 58 |
| CSR-60 | 14 | 18 | 66 | Pant sankar dhan 17 | 11 | 31 | 49 |
| Dobeja-1 | 8 | 14 | 47 | Parimala kalvi | 14 | 22 | 66 |
| Gonrra bhog | 14 | 19 | 57 | Parijat | 13 | 25 | 58 |
| Govind | 12 | 20 | 56 | PR 118 | 8 | 18 | 43 |
| Gowrisanna | 10 | 28 | 56 | Pusa basmati 1 | 13 | 21 | 67 |
| Himalya 799 | 12 | 31 | 61 | Pusa Sugandh 3 | 13 | 34 | 58 |
| HLR-108 | 13 | 34 | 58 | Pusa Sugandh 4 | 14 | 18 | 66 |
| HLR-142 | 13 | 34 | 58 | Ram Jawain 100 | 6 | 28 | 55 |
| HPR 2083 | 12 | 20 | 56 | Ranbir basmati | 14 | 18 | 66 |
| HPR-2178 | 11 | 17 | 32 | Sadabahar | 10 | 27 | 66 |
| HR-12 | 5 | 19 | 28 | Samleshwari | 4 | 17 | 39 |
| IC356437 | 14 | 18 | 66 | Sanna mullare | 6 | 15 | 39 |
| Indira sona | 14 | 27 | 68 | Sathia -2 | 10 | 19 | 60 |
| Indrayani | 8 | 19 | 76 | Satti | 13 | 34 | 58 |
| INRC 779 | 6 | 15 | 39 | Shiva | 8 | 19 | 35 |
| IR 64 | 9 | 10 | 53 | Superbasmati | 14 | 18 | 66 |
| IRAT-144 | 14 | 24 | 55 | Suphala | 13 | 33 | 56 |
| IRBB 55 | 17 | 30 | 76 | T23 | 14 | 18 | 66 |
| IRBB-13 | 13 | 34 | 58 | Tadukan | 3 | 13 | 29 |
| IRBB-4 | 13 | 42 | 89 | Taipei-309 | 6 | 15 | 39 |
| Jatto | 15 | 22 | 66 | Thule ate | 17 | 21 | 57 |
| Kari kantiga | 14 | 18 | 66 | Tilak chandan | 14 | 18 | 66 |
| Kariya | 14 | 18 | 66 | Tiyun | 10 | 21 | 50 |
| Kasturi | 14 | 18 | 66 | V L Dhan | 6 | 6 | 34 |
| Kavali kannu | 2 | 15 | 38 | Vanasurya | 6 | 15 | 39 |
| Kulanji pille | 10 | 23 | 45 | Varalu | 10 | 27 | 66 |
| LD-43 (HLR-144) | 15 | 15 | 55 | Varun dhan | 7 | 13 | 47 |

and 113 identical residues, respectively. The Pi54 protein showed RMSD value between 3 and 4.

## Interaction of Pi54 and Avr-Pi54 proteins

Top ten poses of the interactions were obtained by using Discovery studio 2.0. The best pose of interaction was selected depending on the Z-dock score. Total 59 proteins showed significant interaction with Avr-Pi54 proteins. The interaction images generated are given in Fig 4 and S3 Fig. There were 13 proteins which belonged to rice lines HR12, Mesebetta, Shiva, Mahamaya, Parijat, Malviya Dhan, Ram Jawain 100, Lalankanda, Ranbir basmati, Sathia-2, Satti,

**Table 4. Energy parameters (kcal/mol) of the Pi54 proteins calculated by the CHARMm force field.**

| Rice lines | Potential Energy | Van der waals Energy | Electrostatic Energy | Total Energy |
|---|---|---|---|---|
| Tetep | -17820.86 | -2601.34 | -16611.47 | -37033.67 |
| Acharmita | -19734.10 | -2938.24 | -18377.58 | -41049.93 |
| Basmati 386 | -19706.05 | -2909.36 | -18377.58 | -40993.01 |
| Belgaum basmati | -19639.22 | -2909.36 | -18377.58 | -40926.17 |
| Bidarlocal-2 | -19693.89 | -2909.36 | -18336.75 | -40940.01 |
| Budda | -19520.04 | -2909.36 | -18236.06 | -40665.47 |
| Casbatta | -19520.04 | -2887.16 | -18236.06 | -40643.27 |
| Chiti zhini | -19514.89 | -2887.16 | -18236.06 | -40638.12 |
| CN-1789 | -19514.89 | -2887.16 | -18236.06 | -40638.12 |
| CSR 10 | -19514.89 | -2887.16 | -18155.40 | -40557.45 |
| CSR-60 | -19514.89 | -2885.53 | -18155.40 | -40555.82 |
| Dobeja-1 | -19392.08 | -2864.14 | -18089.12 | -40345.36 |
| Gonrra bhog | -18685.14 | -2864.14 | -17509.05 | -39058.35 |
| Govind | -18685.14 | -2864.14 | -17309.29 | -38858.58 |
| Gowrisanna | -18685.14 | -2864.14 | -17309.29 | -38858.58 |
| Himalya 799 | -18685.14 | -2864.14 | -17309.29 | -38858.58 |
| HLR-108 | -18685.14 | -2864.14 | -17309.29 | -38858.58 |
| HLR-142 | -18685.14 | -2864.14 | -17309.29 | -38858.58 |
| HPR 2083 | -18685.14 | -2864.14 | -17309.29 | -38858.58 |
| HPR-2178 | -18685.14 | -2864.14 | -17309.29 | -38858.58 |
| HR-12 | -9608.120 | -1391.91 | -9103.56 | -20103.61 |
| IC356437 | -18685.14 | -2864.14 | -17309.29 | -38858.58 |
| Indira sona | -18685.14 | -2864.14 | -17309.29 | -38858.58 |
| Indrayani | -18685.14 | -2864.14 | -17309.29 | -38858.58 |
| INRC 779 | -18685.14 | -2864.14 | -17309.29 | -38858.58 |
| IR 64 | -18685.14 | -2864.14 | -17309.29 | -38858.58 |
| IRAT-144 | -18685.14 | -2864.14 | -17309.29 | -38858.58 |
| IRBB 55 | -18685.14 | -2839.64 | -17309.29 | -38834.08 |
| IRBB-13 | -18651.39 | -2809.70 | -17309.29 | -38770.38 |
| IRBB-4 | -18425.21 | -2747.20 | -17027.72 | -38200.14 |
| Jatto | -18261.65 | -2699.01 | -17022.62 | -37983.29 |
| Kari kantiga | -17820.86 | -2699.01 | -16611.47 | -37131.34 |
| Kariya | -17792.63 | -2699.01 | -16597.36 | -37089.02 |
| Kasturi | -17792.63 | -2648.10 | -16511.46 | -36952.19 |
| Kavali kannu | -9639.09 | -1391.10 | -9034.47 | -20065.48 |
| Kulanji pille | -17792.63 | -2641.98 | -16511.46 | -36946.07 |
| LD-43 (HLR-144) | -17723.68 | -2641.98 | -16511.46 | -36877.12 |
| Lalnakanda | -9627.86 | -1391.91 | -9034.48 | -20054.26 |
| Mahamaya | -17657.51 | -2641.98 | -16458.20 | -36757.69 |
| Malviya dhan | -9627.86 | -1375.08 | -9034.47 | -20037.43 |
| Mesebatta | -9627.86 | -1375.08 | -9034.47 | -20037.43 |
| Mote bangarkaddi | -17657.51 | -2641.98 | -16448.72 | -36748.21 |
| MTU 4870 | -17657.51 | -2608.89 | -16448.72 | -36715.12 |
| MTU-1061 | -17657.51 | -2601.34 | -16448.72 | -36707.57 |
| ND 118 | -17646.81 | -2589.72 | -16448.72 | -36685.25 |
| Orugallu | -17155.67 | -2556.39 | -15947.23 | -35659.29 |
| Pant sankar dhan 1 | -16146.68 | -2425.20 | -15026.83 | -33598.73 |

*(Continued)*

**Table 4.** (Continued)

| Rice lines | Potential Energy | Van der waals Energy | Electrostatic Energy | Total Energy |
|---|---|---|---|---|
| Pant sugandh dhan17 | -16096.46 | -2421.34 | -14874.21 | -33392.02 |
| Parijat | -9627.86 | -1375.08 | -9021.24 | -20024.19 |
| Parimala kalvi | -15682.81 | -2392.41 | -14519.86 | -32595.08 |
| PR 118 | -13556.46 | -2012.44 | -12629.43 | -28198.34 |
| Pusa basmati 1 | -13408.05 | -1990.56 | -12439.45 | -27838.06 |
| Pusa Sugandh 3 | -13408.05 | -1990.56 | -12439.45 | -27838.06 |
| Pusa Sugandh 4 | -13135.24 | -1927.95 | -12299.06 | 27362.26 |
| Ram Jawain 100 | -9595.74 | -1375.08 | -9021.24 | -19992.06 |
| Ranbir basmati | -8884.29 | -1310.59 | -9021.24 | -19216.13 |
| Sadabahar | -13135.24 | -1927.95 | -12299.06 | -27362.26 |
| Samleshwari | -13135.24 | -1927.95 | -12299.06 | -27362.26 |
| Sanna mullare | -13135.24 | -1927.95 | -12299.06 | -27362.26 |
| Sathia -2 | -7249.56 | -1212.63 | -8334.84 | -16797.02 |
| Satti | -6196.61 | -914.87 | -5776.11 | -12887.58 |
| Shiva | -6196.61 | -914.87 | -5776.11 | -12887.58 |
| Superbasmati | -6191.19 | -890.55 | -5762.24 | -12843.98 |
| Suphala | -13135.24 | -1927.95 | -12299.06 | -27362.26 |
| T23 | -6191.19 | -854.72 | -5762.24 | -12808.16 |
| Tadukan | -13135.24 | -1927.95 | -12299.06 | -27362.26 |
| Taipei-309 | -6172.576 | -854.72 | -5744.16 | -12771.46 |
| Thule ate | -13135.24 | -1927.95 | -12299.06 | -27362.26 |
| Tilak chandan | -13135.24 | -1927.95 | -12299.06 | -27362.26 |
| Tiyun | -13135.24 | -1927.95 | -12299.06 | -27362.26 |
| V L Dhan | -12548.91 | -1745.78 | -11742.05 | -26036.73 |
| Vanasurya | -12128.76 | -1717.66 | -11500.36 | -25346.78 |
| Varalu | -11999.71 | -1684.68 | -11297.70 | -24982.09 |
| Varun dhan | -9639.092 | -1607.28 | -9470.56 | -20716.93 |

Superbasmati and TP-309 did not interact with Avr-Pi54. For the interaction of Pi54 and Avr-Pi54 proteins, the H bond length less than 4Å was considered significant. In the cases of the proteins that did not show significant interaction, there were no H-bonds found within 4Å range and large separation was observed between these proteins pairs. The intermolecular interactions of the Pi54 proteins and the Avr-Pi54 protein showed that residues of the LRR domain of the Pi54 proteins participated in the interaction (Table 6). The intermolecular H-bonds were considered for the interaction analysis. More number of H-bonds was formed by the proteins of Casbatta, Gowrisanna and HLR-142 rice lines than the Pi54[tetep] during the interaction.

## Binding free energy of Pi54: Avr-Pi54 proteins

The calculated binding energies for Pi54-AvrPi54 protein complexes for all protein pairs are given in the Table 7. The binding energy was found zero in cases where Pi54 protein did not interact with AvrPi54. There were 16 complexes with lesser binding energy than the Pi54[tetep] protein, which involved proteins from the rice lines Casebatta, Tadukan, V L dhan, Varun dhan, Govind, Acharmita, Kavalikannu, HPR-2083, Budda, Jatto, MTU-4870, Dobeja-1, CN-1789, Indira sona, Kulanji pille and Mote bangarkaddi cultivars. These proteins thus have

**Table 5. The TM-score and RMSD values of alignment of Pi54 proteins to the Pi54$^{tetep}$ protein.**

| Rice lines | TM-score (<5Å) | Identical residues (<5 Å) | RMSD (Å) | Rice lines | TM-score (<5Å) | Identical residues (<5 Å) | RMSD (Å) |
|---|---|---|---|---|---|---|---|
| Acharmita | 0.19 | 26 | 3.44 | Mahamaya | 0.33 | 56 | 3.37 |
| Basmati 386 | 0.24 | 44 | 3.25 | Malviya dhan | 0.31 | 50 | 3.63 |
| Belgaum basmati | 0.21 | 41 | 3.13 | Mesebatta | 0.28 | 47 | 3.63 |
| Bidarlocal-2 | 0.22 | 38 | 3.36 | Mote bangarkaddi | 0.32 | 47 | 3.46 |
| Budda | 0.23 | 31 | 3.34 | MTU 4870 | 0.34 | 61 | 3.65 |
| Casbatta | 0.20 | 40 | 3.27 | MTU-1061 | 0.33 | 51 | 3.4 |
| Chiti zhini | 0.30 | 46 | 3.58 | ND 118 | 0.30 | 62 | 3.34 |
| CN-1789 | 0.31 | 56 | 3.68 | Orugallu | 0.61 | 183 | 3.32 |
| CSR 10 | 0.34 | 64 | 3.57 | Pant sankar dhan 1 | 0.28 | 47 | 3.63 |
| CSR-60 | 0.21 | 41 | 3.13 | Pant sugandh dhan 17 | 0.34 | 67 | 3.59 |
| Dobeja-1 | 0.28 | 41 | 3.43 | Parijat | 0.341 | 66 | 3.58 |
| Gonrra bhog | 0.31 | 50 | 3.73 | Parimala kalvi | 0.30 | 46 | 3.58 |
| Govind | 0.36 | 62 | 3.63 | PR 118 | 0.21 | 36 | 3.54 |
| Gowrisanna | 0.33 | 51 | 3.81 | Pusa 33 | 0.26 | 41 | 3.39 |
| Himalya 799 | 0.27 | 42 | 3.59 | Pusa basmati 1 | 0.23 | 36 | 3.44 |
| HLR-108 | 0.34 | 64 | 3.57 | Pusa Sugandh 3 | 0.28 | 47 | 3.63 |
| HLR-142 | 0.34 | 64 | 3.57 | Pusa Sugandh 4 | 0.20 | 41 | 3.13 |
| HPR 2083 | 0.47 | 124 | 3.3 | Ram Jawain 100 | 0.44 | 104 | 3.27 |
| HPR-2178 | 0.27 | 51 | 3.55 | Ranbir basmati | 0.21 | 41 | 3.13 |
| HR-12 | 0.21 | 40 | 3.27 | Sadabahar | 0.34 | 66 | 3.58 |
| IC356437 | 0.21 | 41 | 3.13 | Samleshwari | 0.22 | 41 | 3.34 |
| Indira sona | 0.31 | 48 | 3.54 | Sanna mullare | 0.28 | 47 | 3.63 |
| Indrayani | 0.37 | 69 | 3.55 | Sathia -2 | 0.35 | 78 | 3.55 |
| INRC 779 | 0.28 | 47 | 3.63 | Satti | 0.34 | 64 | 3.57 |
| IR 64 | 0.34 | 68 | 3.34 | Shiva | 0.20 | 29 | 3.34 |
| IRAT-144 | 0.46 | 113 | 3.33 | Superbasmati | 0.21 | 41 | 3.13 |
| IRBB 55 | 0.25 | 38 | 3.32 | Suphala | 0.34 | 66 | 3.58 |
| IRBB-13 | 0.28 | 47 | 3.63 | T23 | 0.20 | 41 | 3.13 |
| IRBB-4 | 0.24 | 36 | 3.55 | Tadukan | 0.21 | 35 | 3.39 |
| Jatto | 0.27 | 47 | 3.63 | Taipei-309 | 0.28 | 47 | 3.63 |
| Kari kantiga | 0.21 | 41 | 3.13 | Thule ate | 0.31 | 51 | 3.61 |
| Kariya | 0.21 | 41 | 3.13 | Tilak chandan | 0.21 | 41 | 3.13 |
| Kasturi | 0.21 | 41 | 3.13 | Tiyun | 0.34 | 58 | 3.61 |
| Kavali kannu | 0.20 | 37 | 3.13 | V L Dhan | 0.17 | 26 | 3.1 |
| Kulanji pille | 0.37 | 58 | 3.63 | Vanasurya | 0.28 | 47 | 3.63 |
| LD-43 (HLR-144) | 0.26 | 38 | 3.59 | Varalu | 0.34 | 66 | 3.58 |
| Lalnakanda | 0.21 | 41 | 3.13 | Varun dhan | 0.30 | 53 | 3.5 |

stronger interaction than the Pi54$^{tetep}$ with the cognate partner AvrPi54 protein of the *M. oryzae*.

## Molecular dynamic simulation

Simulation is used for minimization of the energy and assessed the stability of the docked proteins complex. The complex acquired a stable conformation during the simulation trajectory, after deviating for about 1 Å in the first ns. The RMSD and RMSF for each residue of the complex were computed. RMSD is the global measure of fluctuations and is used to access the

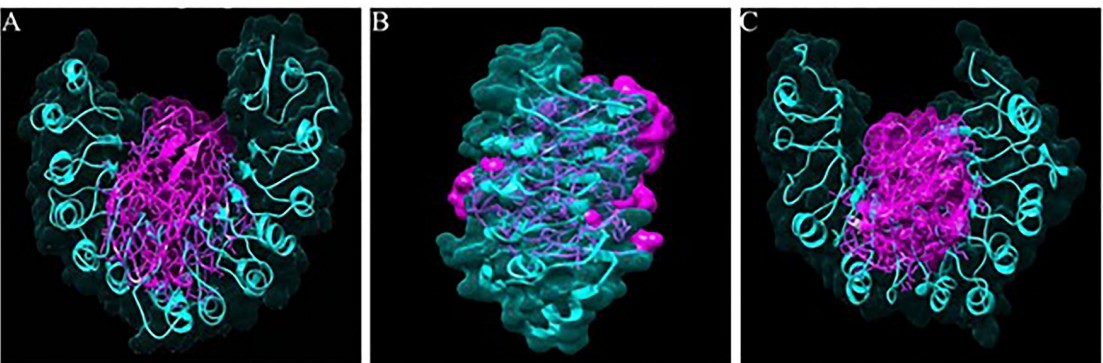

**Fig 4. Docking images showing interaction of Avr-Pi54 and Pi54 proteins.** Interaction with AvrPi54 with R proteins from rice lines, A-Tetep, B- Casbatta, C- HPR -2083.

dynamic stability of the complex and RMSF is the time-average of RMSD for each residue. The number of Hydrogen bonds throughout simulations between the protein pair exhibited nominal change (Fig 5) showing that the complex remained stable during the rest of trajectory. Most of the peaks of RMSF plot possess a value between 0.1 and 0.4 nm. Thus favourable changes were achieved in the simulation process of docked AvrPi54 protein and the Pi54 proteins.

## Discussion

Natural variations harboured in the wild species, landraces and traditional varieties are the source of many agronomically important traits that furnish genes conferring resistance to diseases and pests and adaptation to environmental stresses. Disease resistance alleles from crop gene pool can be introgressed into high yielding varieties to further improve their capacity to overcome the challenges of pathogen attack [34]. Rice blast disease can be effectively managed by the utilisation of *R* gene which can be taken from the resistant varieties to develop high yielding varieties either by breeding or in raising transgenics [35]. The copy number variation, nucleotide diversity, substitutions at the *R* genes loci are sources of allelic variation [17,36]. *R* genes may harbour high or low levels of polymorphism [37]. The genome wide SNPs (Single Nucleotide Polymorphism) in these genes were also discovered in multiple rice genotypes [38]. When our docking results were compared to the phenotyping results from earlier studies [39], it was found that the interacting proteins belonged to the resistant rice line while the non-interacting proteins were from the susceptible rice lines (Table 8).

*Pi54* alleles contain polymorphism at the nucleotide level and are intermediately diversified due to the evolutionary pressure from pathogen side. The transition type SNPs occur more frequently than the transversions in these sequences. The presence of InDels of variable sizes and SNPs at various sites result into amino acids substitutions at the protein level [39,40]. A great variation in the lengths of these proteins is seen which is due to the presence of InDels in the coding region. The changes in amino acid residues and size of the proteins have led to the variations in their molecular weights. These proteins showed a very little difference in average composition of amino acids. The average leucine percentage of the Pi54 proteins was similar to that of the leucine percentage of Pi54^tetep protein. Resistant Pi54 proteins contained nearly 0.7% higher leucine content than the susceptible proteins. Many variations were obtained in their physiochemical properties. Most of the proteins were acidic in nature as indicated by the pI values. GRAVY index was found positive in all the proteins of susceptible alleles except the protein derived from rice line HR-12. It was negative for 14 resistant proteins most of which

**Table 6.  Intermolecular interaction of the Pi54 proteins with Avr-Pi54 protein.**

| Rice lines | LRR region (bp) | Interacting residues from LRR region | H-bond length (Å) | Interacting atoms | Total numbers of interaction |
|---|---|---|---|---|---|
| Tetep | 267–311 | LEU297-THR32 | 1.00 | HN-O | 19 |
| Acharmita | 112–156 | THR133 -ALA129 | 1.20 | HN-O | 15 |
| Basmati 386 | 372–416 | THR389—THR112 | 1.13 | HN-O | 11 |
| Belgaum basmati | 293–337 | GLY312—ILE7 | 1.95 | HN-O | 17 |
| Bidarlocal-2 | 112–156 | SER150—GLU31 | 1.07 | HN-O | 19 |
| Budda | 150–194 | ASN175—GLY56 | 1.09 | HN-O | 12 |
| Casbatta | 130–157 | GLU146—ILE10 | 1.44 | HN-O | 23 |
| Chiti zhini | 313–357 | CYS345 –LEU11 | 1.45 | HN-O | 19 |
| CN-1789 | 200–244 | ALA219—ILE19 | 1.33 | HN-O | 15 |
| CSR 10 | 313–357 | LYS325—ALA34 | 1.36 | HN-O | 13 |
| CSR-60 | 293–337 | THR310 –ILE10 | 1.38 | HN-O | 14 |
| Dobeja-1 | 155–199 | ALA171—ILE7 | 1.88 | HN-O | 10 |
| Gonrra bhog | 293–337 | LEU312 –SER9 | 1.38 | HN-O | 11 |
| Govind | 240–236 | SER230 –TYR124 | 1.10 | HN-O | 18 |
| Gowrisanna | 192–235 | ILE229—MET1 | 1.49 | HN-O | 21 |
| Himalya 799 | 313–357 | SER336—ALA32 | 2.36 | HN-O | 13 |
| HLR-108 | 293–337 | TYR318—ALA14 | 1.34 | HN-O | 14 |
| HLR-142 | 293–337 | THR295– GLU127 | 1.13 | HN-O | 21 |
| HPR 2083 | 267–311 | LYS285—ALA34 | 1.14 | HN-O | 12 |
| HPR-2178 | 258–304 | CYS266 –THR13 | 1.46 | HN-O | 10 |
| IC356437 | 293–337 | ARG296—THR37 | 1.48 | HN-O | 14 |
| Indira sona | 313–357 | THR329 –SER9 | 1.35 | HN-O | 16 |
| Indrayani | 313–357 | ILE345 –ILE8 | 1.66 | HN-O | 12 |
| INRC 779 | 181–225 | SER224—ARG33 | 2.03 | HN-O | 17 |
| IR 64 | 130–174 | ASN149—GLY56 | 1.90 | HN-O | 13 |
| IRAT-144 | 267–311 | GLU279 –GLN2 | 1.90 | HN-O | 15 |
| IRBB 55 | 392–436 | ALA401—ILE8 | 1.90 | HN-O | 18 |
| IRBB-13 | 293–337 | ILE318 –ILE7 | 1.64 | HN-O | 14 |
| IRBB-4 | 398–442 | MET425—VAL59 | 2.05 | HN-O | 17 |
| Jatto | 157–201 | LYS199—PRO60 | 1.94 | HN-O | 12 |
| Kari kantiga | 293–337 | LYS319—LYS36 | 1.92 | HN-O | 15 |
| Kariya | 293–337 | GLU326—THR37 | 1.48 | HN-O | 10 |
| Kasturi | 293–337 | SER323—ARG33 | 2.35 | HN-O | 17 |
| Kavali kannu | 112–156 | PRO137—THR111 | 1.21 | HN-O | 14 |
| Kulanji pille | 253–297 | ARG296—ALA129 | 1.12 | HN-O | 18 |
| LD-43 (HLR-144) | 313–357 | LEU327 –SER9 | 1.13 | HN-O | 17 |
| Mote bangarkaddi | 181–225 | ALA189—ILE20 | 1.41 | HN-O | 12 |
| MTU 4870 | 296–340 | GLU299—ILE10 | 1.41 | HN-O | 13 |
| MTU-1061 | 293–337 | VAL298—SER9 | 1.31 | HN-O | 12 |
| ND 118 | 183–227 | VAL201—SER9 | 1.42 | HN-O | 15 |
| Orugallu | 267–311 | LYS275—CYS35 | 1.43 | HN-O | 16 |
| Pant sankar dhan 1 | 293–337 | GLN326—ASN135 | 0.98 | HN-O | 12 |
| Pant sugandh dhan 17 | 266–309 | CYS303—GLN2 | 1.03 | HN-O | 10 |
| Parimala kalvi | 313–357 | TRP331 –TYR124 | 1.12 | HN-O | 17 |
| PR 118 | 112–156 | ALA148—LEU30 | 1.53 | HN-O | 18 |
| Pusa basmati 1 | 313–357 | ILE339—ALA5 | 1.53 | HN-O | 13 |
| Pusa Sugandh 3 | 293–337 | ILE312 –ILE7 | 1.60 | HN-O | 16 |
| Pusa Sugandh 4 | 293–337 | TYR298—THR28 | 2.16 | HN-O | 14 |
| Sadabahar | 313–357 | GLY332—ILE7 | 1.76 | HN-O | 15 |

*(Continued)*

**Table 6.** (Continued)

| Rice lines | LRR region (bp) | Interacting residues from LRR region | H-bond length (Å) | Interacting atoms | Total numbers of interaction |
|---|---|---|---|---|---|
| Samleshwari | 112–156 | GLU170 –GLN2 | 1.79 | HN-O | 18 |
| Sanna mullare | 181–225 | ILE212 –ILE8 | 1.86 | HN-O | 14 |
| Suphala | 313–357 | ILE345– GLN2 | 1.80 | HN-O | 11 |
| Tadukan | 130–174 | CYS145 –LEU11 | 1.57 | HN-O | 16 |
| Thule ate | 181–225 | GLU191—ALA51 | 1.58 | HN-O | 15 |
| Tilak chandan | 293–337 | CYS301 –THR13 | 1.59 | HN-O | 14 |
| Tiyun | 290–309 | THR305—THR4 | 1.31 | HN-O | 17 |
| V L Dhan | 112–156 | SER144—ASN50 | 1.10 | HN-O | 17 |
| Vanasurya | 181–225 | SER224—ALA32 | 1.05 | HN-O | 15 |
| Varalu | 313–357 | ILE321 –GLN2 | 1.88 | HN-O | 17 |
| Varun dhan | 250–294 | GLY255—ASN50 | 1.12 | HN-O | 10 |

**Table 7. The binding energy for interaction of AvrPi54 protein with the Pi54 proteins pairs.**

| Rice lines | Binding Energy | Rice lines | Binding Energy |
|---|---|---|---|
| Pi54 (Tetep) | -3835.00 | Kariya | -2764.13 |
| Acharmita | -4105.00 | Kasturi | -2764.13 |
| Basmati 386 | -3643.58 | Kavali kannu | -4015.00 |
| Belgaum basmati | -2764.13 | Kulanji pille | -3850.77 |
| Bidarlocal-2 | -3295.00 | LD-43 (HLR-144) | -1802.77 |
| Budda | -3980.19 | Mote bangarkaddi | -3850.77 |
| Casbatta | -4302.89 | MTU 4870 | -3980.19 |
| Chiti zhini | -1762.09 | MTU-1061 | -3588.15 |
| CN-1789 | -3850.77 | ND 118 | -3745.00 |
| CSR 10 | -2764.13 | Orugallu | -3682.71 |
| CSR-60 | -2764.13 | Pant sankar dhan 1 | -2009.77 |
| Dobeja-1 | -3925.00 | Pant sankar dhan 17 | -3817.33 |
| Gonrra bhog | -2959.16 | Parimala kalvi | -1318.96 |
| Govind | -4202.68 | PR 118 | -3385.00 |
| Gowrisanna | -2764.13 | Pusa basmati 1 | -1968.48 |
| Himalya 799 | -3778.33 | Pusa Sugandh 3 | -2009.77 |
| HLR-108 | -2552.63 | Pusa Sugandh 4 | -2764.13 |
| HLR-142 | -2419.67 | Sadabahar | -2959.16 |
| HPR 2083 | -3986.22 | Samleshwari | -3205.00 |
| HPR-2178 | -3115.00 | Sanna mullare | -3565.00 |
| IC356437 | -2764.13 | Suphala | -1968.48 |
| Indira sona | -3850.77 | Tadukan | -4302.89 |
| Indrayani | -3790.43 | Thule ate | -1968.48 |
| INRC 779 | -3655.00 | Tilak chandan | -2764.13 |
| IR 64 | -3810.73 | Tiyun | -3512.26 |
| IRAT-144 | -1968.48 | V L Dhan | -4291.66 |
| IRBB 55 | -1289.09 | Vanasurya | -3475.00 |
| IRBB-13 | -2009.77 | Varalu | -2939.47 |
| IRBB-4 | -840.09 | Varun dhan | -4240.00 |
| Jatto | -3980.19 | Varun dhan | -613.72 |
| Kari kantiga | -2764.13 | | |

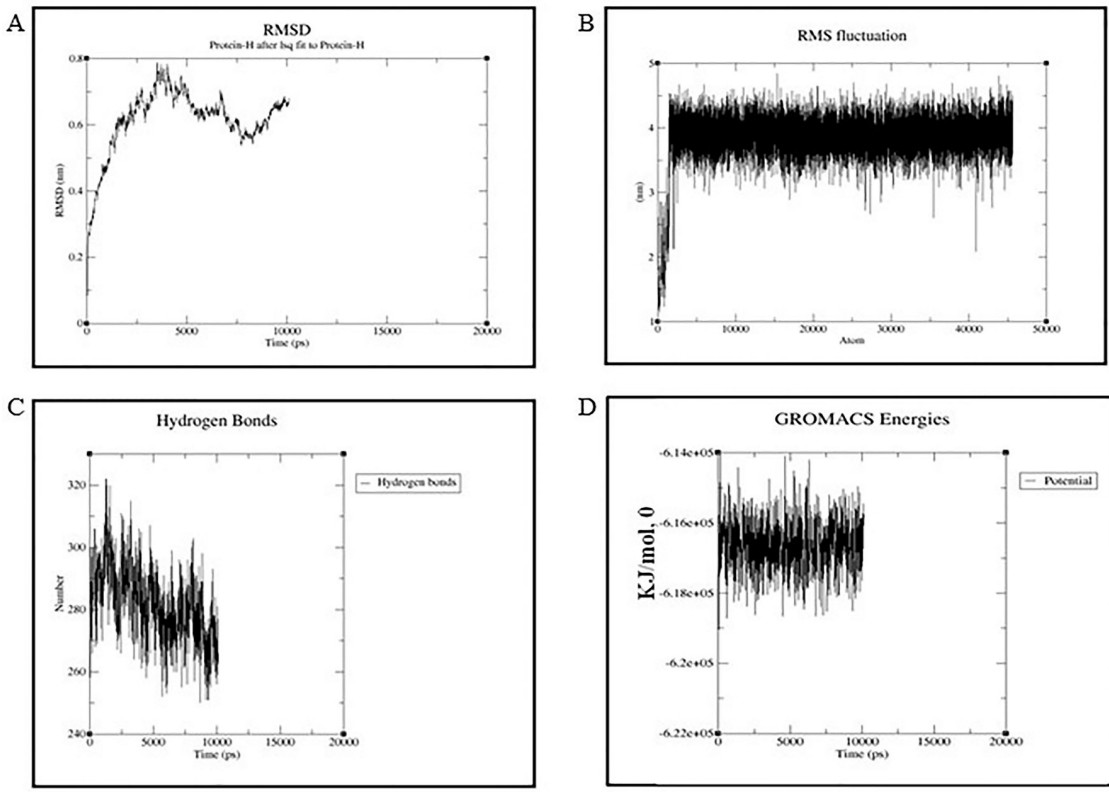

**Fig 5. Molecular dynamics simulation of AvrPi54-Pi54<sup>tetep</sup> protein complex.** The complex acquired a stable conformation during the simulation trajectory, after deviating for about 1 Å in the first ns. Most of the peaks of RMSF plot possess a value between 0.1 and 0.4 nm. Thus favourable changes were achieved in the simulation process of docked AvrPi54 protein and the Pi54 proteins. **(A)** The Root Mean Square Deviation (RMSD), **(B)** Root Mean Square Fluctuation (RMSF), **(C)** Hydrogen bonds, and **(D)** Gyration energy plots.

were found to bind more strongly with AvrPi54 than Pi54<sup>tetep</sup> protein. The negative (−) GRAVY scores indicate that these proteins are hydrophilic. The low GRAVY index of resistance proteins infers that these hydrophilic proteins have more residues available for the formation of intermolecular hydrogen bonds while interacting with the AvrPi54 protein. These forces are significant in association and stability of protein complexes [41,42]. Therefore, polymorphism in the nucleotide sequence of the alleles has led to the variation in the amino acid sequence, leading to the differences in the physiochemical properties of these proteins.

The AvrPi54 protein was found to directly interact with the LRR domain of the resistant Pi54 proteins. The direct interaction between Pi54 and AvrPi54 proteins has earlier been confirmed *in vitro* by using the Yeast-two-hybrid analysis and *in planta* Tobacco leaf infiltration assay [20]. Non-synonymous substitutions were observed in LRR region of Pi54 gene. LRR domains are known to be located at the carboxy termini of plant resistance proteins. The domain acquires barrel-like structure lined with parallel β-sheets in the inner surface and α-helical structures in the remaining region. These structural units are arranged in the manner so that the protein acquires a non- globular shape similar to the horse shoe structure and is responsible for the protein- binding functions of the proteins [43]. In our analysis, the LRR domain was more conserved in the proteins from resistant cultivars than the susceptible ones. The interaction of AvrPi54 protein might have been interrupted with the susceptible alleles due to the amino acids substitution in the LRR region which seems to prevent the proper

**Table 8. Categorisation of the Pi54 proteins into resistant and susceptible lines.**

| S. No. | Rice line | Phenotype | S. No. | Rice line | Phenotype |
|---|---|---|---|---|---|
| 1 | Acharmati | R | 35 | Malviya dhan | S |
| 2 | Basmati 386 | R | 36 | Mesebatta | S |
| 3 | Belgaum basmati | R | 37 | Mote bangarkaddi | R |
| 4 | Bidarlocal-2 | R | 38 | MTU-1061 | R |
| 5 | Budda | R | 39 | Orugallu | R |
| 6 | Chiti zhini | R | 40 | Pant sankar dhan 1 | R |
| 7 | CN-1789 | R | 41 | Pant sugandh dhan 17 | R |
| 8 | CO-39 | S | 42 | Parijat | S |
| 9 | CSR 10 | R | 43 | Parimala kalvi | R |
| 10 | CSR-60 | R | 44 | Pi54 (Tetep) | R |
| 11 | Gonrra bhog | R | 45 | PR 118 | R |
| 12 | Gowrisanna | R | 46 | Pusa basmati 1 | S |
| 13 | Himalya 799 | R | 47 | Pusa Sugandh 3 | R |
| 14 | HLR-108 | R | 48 | Ram Jawain 100 | S |
| 15 | HLR-142 | R | 49 | Ranbir basmati | R |
| 16 | HPR 2083 | R | 50 | Sadabahar | R |
| 17 | HPR-2178 | R | 51 | Samleshwari | R |
| 18 | HR-12 | S | 52 | Sanna mullare | R |
| 19 | Indira sona | R | 53 | Sathia -2 | S |
| 20 | Indrayani | R | 54 | Satti | S |
| 21 | INRC 779 | R | 55 | Shiva | S |
| 22 | IR 64 | R | 56 | Superbasmati | S |
| 23 | IRAT-144 | R | 57 | Suphala | R |
| 24 | IRBB 55 | R | 58 | T23 | S |
| 25 | IRBB-13 | R | 59 | Tadukan | R |
| 26 | IRBB-4 | R | 60 | Taipei-309 | S |
| 27 | Jatto | R | 61 | Thule atte | R |
| 28 | Kari kantiga | R | 62 | Tilak chandan | R |
| 29 | Kariya | R | 63 | Tiyun | R |
| 30 | Kavali kannu | S | 64 | V L Dhan 21 | R |
| 31 | Kulanji pille | R | 65 | Vanasurya | R |
| 32 | Lalnakanda | S | 66 | Varalu | R |
| 33 | HLR-144 | R | 67 | Varun dhan | R |
| 34 | Mahamaya | R | | | |

folding of the interacting LRR region for interaction. Mutation in LRR region, which maintains gene-for-gene specificity, may increase or decrease the recognition specificity [44]. Several studies have reported the effect of mutation in LRR domain on recognition capability and ligand binding specificity of R proteins. Change in a single amino acid in the β-strand region of the LRRs of polygalacturonase-inhibiting proteins confers a new recognition capability and increases ligand specificity in *Phaseolus vulgaris* [45]. A mutation that substitutes the amino acid glutamate to lysine within the LRR domain of a resistance gene RPS5 of Arabidopsis causes reduction in the resistance potential of several *R* genes that conferred resistance against multiple bacterial and downy mildew diseases [46]. Similarly, in the case of rice-*M. oryzae* pathosystem, a single amino acid change in the xxLxLxx motif of R proteins altered the surface through which they interact with their cognate Avr proteins [47].

The folding of the protein to its native structure is driven by various forces such as Hydrogen bonds that pack the helices and strands; Vander Waals interactions that tightly pack the atoms within a folded protein and the backbone angle preferences [48]. The secondary structure of the Pi54 proteins contained more number of turns and H-bonds in resistant cultivars. Numbers of α-helices were same in both resistant and susceptible proteins but the numbers of β-strands were found higher in case of resistance proteins. These factors could be accountable for greater stability of structures of resistance alleles in comparison with the susceptible alleles and also proper folding of the LRR region into protein interacting domain.

The occurrence of SNPs and InDels has created differences in the protein structures which gave rise to the differences in the interaction or interacting strength of Pi54 proteins with AvrPi54 protein. In this study, we found that some of the Pi54 proteins were structurally not similar to the Pi54^tetep protein. Though the Pi54 proteins had sequence similarity but these have very less residual similarity in three dimensional spatial arrangements. The similarity in the LRR region of Pi54 proteins of resistance alleles with the Pi54^tetep protein could be one of the reasons to allow successful interaction with the AvrPi54 protein. Binding energy for the Pi54 and Avr-Pi54 proteins also varied due to the structure variation among these proteins. The binding energy is a dependent variable on the global free minimum energy of the proteins. More number of H-bonds was found to minimize the binding energy of proteins. Out of 59 resistant proteins, only 15 resistant proteins from the land races: the Casebatta, Tadukan, VL Dhan, Varun dhan, Govind, Acharmita, HPR-2083, Budda, Jatto, MTU-4870, Dobeja-1, CN-1789, Indira sona, Kulanji pille and Motebangarkaddi were observed to show lower binding free energy with Avr-Pi54 proteins as compared to the Pi54^tetep protein, thus a stronger interaction potential.

## Conclusion

Our studies show that the nucleotide polymorphism in the Pi54 alleles is the cause of variations in physiochemical properties, LRR domain structure, protein structure, global free minimum energy, H-bond and global protein structures of both the resistant and susceptible alleles. These variations also affect the Pi54 and AvrPi54 interactions. The proteins from the resistant land races like Casebatta, Tadukan, VL Dhan, Varun dhan, Govind, Acharmita, HPR-2083, Budda, Jatto, MTU-4870, Dobeja-1, CN-1789, Indira sona, Kulanji pille and Mote bangarkaddi show stronger bonds with AvrPi54 protein than the Pi54^tetep protein. The resistant response would escalate if the interaction of the R and Avr partners is stronger [49]. Therefore, these alleles have more potential than the original resistance allele and can be effectively used for the rice blast resistance breeding program in future.

## Supporting information

**S1 Table. List of *Pi54* alleles used and their source rice lines.**
(PDF)

**S1 Fig. Composition of different amino acids in Pi54 proteins of the alleles cloned from resistant and susceptible rice lines compared with wild type Pi54^tetep protein.**
(PDF)

**S2 Fig. Three dimensional structures of various Pi54 proteins from rice lines.**
(PDF)

**S3 Fig. Docking images showing interaction of AvrPi54 protein and Pi54 proteins from rice lines.**
(PDF)

## Acknowledgments

TRS is thankful to the Department of Biotechnology, Govt. of India for financial help and Department of Science and Technology, Govt of India for JC Bose National Fellowship. CS is thankful to IARI, New Delhi, India for the financial assistance and the staffs of TAC, IASRI and CAS Lab. BKS is thankful to CSIR, New Delhi, India for research grant.

## Author Contributions

**Conceptualization:** Tilak Raj Sharma.

**Formal analysis:** Pankaj Kumar Singh.

**Investigation:** Chiranjib Sarkar.

**Writing – original draft:** Banita Kumari Saklani.

**Writing – review & editing:** Ravi Kumar Asthana, Tilak Raj Sharma.

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
