## [Decision Letter · Decision Letter 0]

28 Jul 2019

PONE-D-19-16056

In silico interaction of Pi54-AvrPi54 proteins advances molecular understanding in the rice-M. oryzae pathosystem

PLOS ONE

Dear Dr. Sharma,

Thank you for submitting your manuscript to PLOS ONE. After careful consideration, we feel that it has merit but does not fully meet PLOS ONE’s publication criteria as it currently stands. Therefore, we invite you to submit a revised version of the manuscript that addresses the points raised during the review process.

ACADEMIC EDITOR: Please insert comments here and delete this placeholder text when finished. Be sure to:

Indicate which changes are required versus recommended for acceptanceAddress any conflicts between the reviewsProvide specific feedback from your evaluation of the manuscript

We would appreciate receiving your revised manuscript by Sep 11 2019 11:59PM. To enhance the reproducibility of your results, we recommend that if applicable you deposit your laboratory protocols in protocols.io, where a protocol can be assigned its own identifier (DOI) such that it can be cited independently in the future. For instructions see: http://journals.plos.org/plosone/s/submission-guidelines#loc-laboratory-protocols

We look forward to receiving your revised manuscript.

Kind regards,

Zonghua Wang, Ph.D.

Academic Editor

PLOS ONE

Journal Requirements:

1. We note that you have stated that you will provide repository information for your data at acceptance. Should your manuscript be accepted for publication, we will hold it until you provide the relevant accession numbers or DOIs necessary to access your data. If you wish to make changes to your Data Availability statement, please describe these changes in your cover letter and we will update your Data Availability statement to reflect the information you provide.

Reviewers' comments:

Reviewer's Responses to Questions

**Comments to the Author**

1. Is the manuscript technically sound, and do the data support the conclusions?

Reviewer #1: Partly

Reviewer #2: Yes

2. Has the statistical analysis been performed appropriately and rigorously? 

Reviewer #1: Yes

Reviewer #2: Yes

3. Have the authors made all data underlying the findings in their manuscript fully available?

Reviewer #1: Yes

Reviewer #2: Yes

4. Is the manuscript presented in an intelligible fashion and written in standard English?

Reviewer #1: No

Reviewer #2: Yes

5. Review Comments to the Author

Reviewer #1: The papers presents the results of the nucleotide polymoephism in the Pi54 alleles which affect the Pi54 and AvrPi54 interaction. The title of the papers should be more specific since many of studies have been done on the R-Avr protein interaction including in the author papers published in 2014,which revealed that the Pi54of interacted with AVR-Pi54 through its non-LRR region, but Pi54 and Pi54rh interacted through CC-non-LRR-LRR regions and non-LRR-LRR regions, and the susceptible Pi54tp protein did not show interaction with AVR-Pi54, respectively. So this papers judged from the title only provides a study is similar to the former’.

The major problem of the papers is that the conclusions in the papers should be verified by experiment, the molecular simulation is not real happened after all. Furthermoer, there are many problems with sentence structure and clause constrution.

So try to set the problem discussed above in more detail!

Reviewer #2: The authors analysed the Pi54 alleles from 72 rice lines by using bioinformatics methods to understand the interaction of Pi54 (R) proteins with AvrPi54 (Avr) protein. And they found that the physiochemical properties of these proteins varied due to the nucleotide level polymorphism. Generally, the authors present a well conducted study. The manuscript is concise and the results are mainly convincing.

Some minor points:

1. Figure 1, it can be hard to distinguish the 13 different amino acids. It may be useful to draw some lines.

2. Figure 2, add the letters to indicate which one is 2A or 2B.

3. It could be better for this manuscript if the authors could do some Y2H to check the interactions between avrpi54 and Pi54 alleles from resistant and susceptible lines. These experimental results could provide stronger evidence to explain the different phenotype

6. PLOS authors have the option to publish the peer review history of their article (what does this mean?). If published, this will include your full peer review and any attached files.

Reviewer #1: No

Reviewer #2: No

---

## [Author Response · Author response to Decision Letter 0]

9 Sep 2019

Reviewer 1 comments 

The papers presents the results of the nucleotide polymorphisms in the Pi54 alleles which affect the Pi54 and AvrPi54 interaction. The title of the papers should be more specific since many of studies have been done on the R-Avr protein interaction including in the author papers published in 2014, which revealed that the Pi54of interacted with AVR-Pi54 through its non-LRR region, but Pi54 and Pi54rh interacted through CC-non-LRR-LRR regions and non-LRR-LRR regions, and the susceptible Pi54tp protein did not show interaction with AVR-Pi54, respectively. So this papers judged from the title only provides a study is similar to the former’.

The major problem of the papers is that the conclusions in the papers should be verified by experiment, the molecular simulation is not real happened after all. Furthermore, there are many problems with sentence structure and clause construction.

So try to set the problem discussed above in more detail!)

Our reply 

Title of the manuscript is changed as per the suggestion of the reviewer that is “Variation in the LRR region of Pi54 protein alters its interaction with the AvrPi54 protein revealed by in silico analysis”.

The experimental proof for the interaction between Avr-Pi54 and Pi54 proteins was already done by Ray et al, 2016 (Front Plant Sci. 2016; 7: 1140). Here, the main objectives of this manuscript was to analyse sequence variation at the LRR regions of those alleles of Pi54 gene isolated from both phenotypically resistant and susceptible rice lines and analysed sequence variation at the LRR region and look at their interaction with AvrPi54 gene. The hypothesis was that the proteins of Pi54 alleles isolated from resistant rice lines should show in silico interaction with the AvrPi54 proteins and the susceptible one should not show any interaction which is akin to the Flor`s gene-for-gene hypothesis (Flor 1972). The interaction analysis revealed that Pi54 protein from resistance lines, showed significant in silico interactions with AvrPi54 protein, while Pi54 from susceptible lines did not interact with its counterpart. Hence, it proves our hypothesis.

We made corrections in the conclusion section of the manuscript in lines 527 and 529. We also corrected many sentences throughout the manuscript, e.g. in lines 56, 71, 77, 83, etc. These changes can be seen in the track change version of the manuscript.

In addition in the revised version of the MS, molecular dynamics simulation analysis was carried out to see whether the interactions between Avr-Pi54 and Pi54 proteins are stable at the thermodynamics level or not, and found that the interaction was stable at the different thermodynamic parameters.

Reviewer 2 comments: 

The authors analysed the Pi54 alleles from 72 rice lines by using bioinformatics methods to understand the interaction of Pi54 (R) proteins with AvrPi54 (Avr) protein. And they found that the physiochemical properties of these proteins varied due to the nucleotide level polymorphism. Generally, the authors present a well conducted study. The manuscript is concise and the results are mainly convincing.

Some minor points:

1. Figure 1, it can be hard to distinguish the 13 different amino acids. It may be useful to draw some lines.

Our reply

The figure 1 has been removed from the main text and instead of this figure a comparative composition of amino acids of the resistant and susceptible proteins with Pi54tetep protein is given, but annexed to supplementary data as figure S1 titled as “Composition of different amino acids in Pi54 proteins of the alleles cloned from resistant and susceptible rice line (A) compared with wild type Pi54tetep protein”.

2. Figure 2, add the letters to indicate which one is 2A or 2B.

Our reply

As suggested the changes have been made. 

3. It could be better for this manuscript if the authors could do some Y2H to check the interactions between avrpi54 and Pi54 alleles from resistant and susceptible lines. These experimental results could provide stronger evidence to explain the different phenotype.

Our reply

The Y2H assay has been already performed between the AvrPi54 and Pi54tetep protein to confirm their interaction by Ray et al, 2016, “Analysis of Magnaporthe oryzae genome reveals a fungal effector, which is able to induce resistance response in transgenic rice line containing resistance gene, Pi54” published in Frontiers in Plant Science journal and this publication has been cited in this manuscript. In this article, authors were identified the AvrPi54 gene and validated the gene using the in vitro as well as the in planta experiments. The significant interaction between the candidates of AvrPi54 and the Pi54 protein was initially begun with the bioinformatics approaches and led the analysis to pinpoint the AvrPi54 gene. Thus we hypothesized that to check whether the phenotypic results of rice blast disease screening experiment, on Pi54 resistant rice line and susceptible rice line would be matched with the results of in silico interaction between AvrPi54 and variants of Pi54 proteins. For that we conducted this study and disrupted interactions were found when variants of Pi54 derived from susceptible rice lines interacted with the AvrPi54 protein, while a strong interaction with AvrPi54 was found in case of Pi54 variants obtained from resistant rice lines. Thus, any of the Pi54 allele could be initially determined for its function through the in silico interaction analysis without doing phenotyping experiment. That is the main objective of this study. The interaction between Pi54-AvrPi54 proteins has been taken as a basis to extend our knowledge to discover the strength of interaction between various allelic Pi54 proteins with the AvrPi54 protein.

---

## [Editor Report · Decision Letter 1]

7 Oct 2019

Variation in the LRR region of Pi54 protein alters its interaction with the AvrPi54 protein revealed by in silico analysis

PONE-D-19-16056R1

Dear Dr. Sharma,

We are pleased to inform you that your manuscript has been judged scientifically suitable for publication and will be formally accepted for publication once it complies with all outstanding technical requirements.

With kind regards,

Zonghua Wang, Ph.D.

Academic Editor

PLOS ONE
---

## [Editor Report · Acceptance letter]

22 Oct 2019

PONE-D-19-16056R1 

Variation in the LRR region of Pi54 protein alters its interaction with the AvrPi54 protein revealed by *in silico* analysis

Dear Dr. Sharma:

I am pleased to inform you that your manuscript has been deemed suitable for publication in PLOS ONE. Congratulations! Your manuscript is now with our production department. 

With kind regards,

on behalf of

Prof. Zonghua Wang 

Academic Editor

PLOS ONE